# The Pseudo-Dimension of Near-Optimal Auctions

**Jamie Morgenstern**[*]
Computer and Information Science
University of Pennsylvania
Philadelphia, PA
jamiemor@cis.upenn.edu

**Tim Roughgarden**
Stanford University
Palo Alto, CA
tim@cs.stanford.edu

## Abstract

This paper develops a general approach, rooted in statistical learning theory, to learning an approximately revenue-maximizing auction from data. We introduce *t-level auctions* to interpolate between simple auctions, such as welfare maximization with reserve prices, and optimal auctions, thereby balancing the competing demands of expressivity and simplicity. We prove that such auctions have small representation error, in the sense that for every product distribution $F$ over bidders' valuations, there exists a $t$-level auction with small $t$ and expected revenue close to optimal. We show that the set of $t$-level auctions has modest pseudo-dimension (for polynomial $t$) and therefore leads to small learning error. One consequence of our results is that, in arbitrary single-parameter settings, one can learn a mechanism with expected revenue arbitrarily close to optimal from a polynomial number of samples.

## 1 Introduction

In the traditional economic approach to identifying a revenue-maximizing auction, one first posits a prior distribution over all unknown information, and then solves for the auction that maximizes expected revenue with respect to this distribution. The first obstacle to making this approach operational is the difficulty of formulating an appropriate prior. The second obstacle is that, even if an appropriate prior distribution is available, the corresponding optimal auction can be far too complex and unintuitive for practical use. This motivates the goal of identifying auctions that are "simple" and yet nearly-optimal in terms of expected revenue.

In this paper, we apply tools from learning theory to address both of these challenges. In our model, we assume that bidders' valuations (i.e., "willingness to pay") are drawn from an unknown distribution $F$. A learning algorithm is given i.i.d. samples from $F$. For example, these could represent the outcomes of comparable transactions that were observed in the past. The learning algorithm suggests an auction to use for future bidders, and its performance is measured by comparing the expected revenue of its output auction to that earned by the optimal auction for the distribution $F$.

The possible outputs of the learning algorithm correspond to some set $\mathcal{C}$ of auctions. We view $\mathcal{C}$ as a design parameter that can be selected by a seller, along with the learning algorithm. A central goal of this work is to identify classes $\mathcal{C}$ that balance representation error (the amount of revenue sacrificed by restricting to auctions in $\mathcal{C}$) with learning error (the generalization error incurred by learning over $\mathcal{C}$ from samples). That is, we seek a set $\mathcal{C}$ that is rich enough to contain an auction that closely approximates an optimal auction (whatever $F$ might be), yet simple enough that the best auction in $\mathcal{C}$ can be learned from a small amount of data. Learning theory offers tools both for rigorously defining the "simplicity" of a set $\mathcal{C}$ of auctions, through well-known complexity measures such as the

---

[*]Part of this work done while visiting Stanford University. Partially supported by a Simons Award for Graduate Students in Theoretical Computer Science, as well as NSF grant CCF-1415460.

pseudo-dimension, and for quantifying the amount of data necessary to identify the approximately best auction from $\mathcal{C}$. Our goal of learning a near-optimal auction also requires understanding the representation error of different classes $\mathcal{C}$; this task is problem-specific, and we develop the necessary arguments in this paper.

## 1.1 Our Contributions

The primary contributions of this paper are the following. First, we show that well-known concepts from statistical learning theory can be directly applied to reason about learning from data an approximately revenue-maximizing auction. Precisely, for a set $\mathcal{C}$ of auctions and an arbitrary unknown distribution $F$ over valuations in $[1, H]$, $O(\frac{H^2}{\epsilon^2} \mathrm{d}_{\mathcal{C}} \log \frac{H}{\epsilon})$ samples from $F$ are enough to learn (up to a $1 - \epsilon$ factor) the best auction in $\mathcal{C}$, where $\mathrm{d}_{\mathcal{C}}$ denotes the *pseudo-dimension* of the set $\mathcal{C}$ (defined in Section 2). Second, we introduce the class of *t-level auctions*, to interpolate smoothly between simple auctions, such as welfare maximization subject to individualized reserve prices (when $t = 1$), and the complex auctions that can arise as optimal auctions (as $t \to \infty$). Third, we prove that in quite general auction settings with $n$ bidders, the pseudo-dimension of the set of $t$-level auctions is $O(nt \log nt)$. Fourth, we quantify the number $t$ of levels required for the set of $t$-level auctions to have low representation error, with respect to the optimal auctions that arise from arbitrary product distributions $F$. For example, for single-item auctions and several generalizations thereof, if $t = \Omega(\frac{H}{\epsilon})$, then for every product distribution $F$ there exists a $t$-level auction with expected revenue at least $1 - \epsilon$ times that of the optimal auction for $F$.

In the above sense, the "$t$" in $t$-level auctions is a tunable "sweet spot", allowing a designer to balance the competing demands of expressivity (to achieve near-optimality) and simplicity (to achieve learnability). For example, given a fixed amount of past data, our results indicate how much auction complexity (in the form of the number of levels $t$) one can employ without risking overfitting the auction to the data.

Alternatively, given a target approximation factor $1 - \epsilon$, our results give sufficient conditions on $t$ and consequently on the number of samples needed to achieve this approximation factor. The resulting sample complexity upper bound has polynomial dependence on $H$, $\epsilon^{-1}$, and the number $n$ of bidders. Known results [1, 8] imply that any method of learning a $(1 - \epsilon)$-approximate auction from samples must have sample complexity with polynomial dependence on all three of these parameters, even for single-item auctions.

## 1.2 Related Work

The present work shares much of its spirit and high-level goals with Balcan et al. [4], who proposed applying statistical learning theory to the design of near-optimal auctions. The first-order difference between the two works is that our work assumes bidders' valuations are drawn from an unknown distribution, while Balcan et al. [4] study the more demanding "prior-free" setting. Since no auction can achieve near-optimal revenue ex-post, Balcan et al. [4] define their revenue benchmark with respect to a set $\mathcal{G}$ of auctions on each input $\mathbf{v}$ as the maximum revenue obtained by any auction of $\mathcal{G}$ on $\mathbf{v}$. The idea of learning from samples enters the work of Balcan et al. [4] through the internal randomness of their partitioning of bidders, rather than through an exogenous distribution over inputs (as in this work). Both our work and theirs requires polynomial dependence on $H, \frac{1}{\epsilon}$: ours in terms of a necessary number of samples, and theirs in terms of a necessary number of bidders; as well as a measure of the complexity of the class $\mathcal{G}$ (in our case, the pseudo-dimension, and in theirs, an analogous measure). The primary improvement of our work over of the results in Balcan et al. [4] is that our results apply for single item-auctions, matroid feasibility, and arbitrary single-parameter settings (see Section 2 for definitions); while their results apply only to single-parameter settings of unlimited supply.[1] We also view as a feature the fact that our sample complexity upper bounds can be deduced directly from well-known results in learning theory — we can focus instead on the non-trivial and problem-specific work of bounding the pseudo-dimension and representation error of well-chosen auction classes.

Elkind [12] also considers a similar model to ours, but only for the special case of single-item auctions. While her proposed auction format is similar to ours, our results cover the far more general

case of arbitrary single-parameter settings and and non-finite support distributions; our sample complexity bounds are also better even in the case of a single-item auction (linear rather than quadratic dependence on the number of bidders). On the other hand, the learning algorithm in [12] (for single-item auctions) is computationally efficient, while ours is not.

Cole and Roughgarden [8] study single-item auctions with $n$ bidders with valuations drawn from independent (not necessarily identical) "regular" distributions (see Section 2), and prove upper and lower bounds (polynomial in $n$ and $\epsilon^{-1}$) on the sample complexity of learning a $(1-\epsilon)$-approximate auction. While the formalism in their work is inspired by learning theory, no formal connections are offered; in particular, both their upper and lower bounds were proved from scratch. Our positive results include single-item auctions as a very special case and, for bounded or MHR valuations, our sample complexity upper bounds are much better than those in Cole and Roughgarden [8].

Huang et al. [15] consider learning the optimal price from samples when there is a single buyer and a single seller; this problem was also studied implicitly in [10]. Our general positive results obviously cover the bounded-valuation and MHR settings in [15], though the specialized analysis in [15] yields better (indeed, almost optimal) sample complexity bounds, as a function of $\epsilon^{-1}$ and/or $H$.

Medina and Mohri [17] show how to use a combination of the pseudo-dimension and Rademacher complexity to measure the sample complexity of selecting a single reserve price for the VCG mechanism to optimize revenue. In our notation, this corresponds to analyzing a single set $\mathcal{C}$ of auctions (VCG with a reserve). Medina and Mohri [17] do not address the expressivity vs. simplicity trade-off that is central to this paper.

Dughmi et al. [11] also study the sample complexity of learning good auctions, but their main results are negative (exponential sample complexity), for the difficult scenario of multi-parameter settings. (All settings in this paper are single-parameter.)

Our work on $t$-level auctions also contributes to the literature on simple approximately revenue-maximizing auctions (e.g., [6, 14, 7, 9, 21, 24, 2]). Here, one takes the perspective of a seller who knows the valuation distribution $F$ but is bound by a "simplicity constraint" on the auction deployed, thereby ruling out the optimal auction. Our results that bound the representation error of $t$-level auctions (Theorems 3.4, 4.1, 5.4, and 6.2) can be interpreted as a principled way to trade off the simplicity of an auction with its approximation guarantee. While previous work in this literature generally left the term "simple" safely undefined, this paper effectively proposes the pseudo-dimension of an auction class as a rigorous and quantifiable simplicity measure.

## 2 Preliminaries

This section reviews useful terminology and notation standard in Bayesian auction design and learning theory.

**Bayesian Auction Design**   We consider *single-parameter settings* with $n$ bidders. This means that each bidder has a single unknown parameter, its *valuation* or willingness to pay for "winning." (Every bidder has value 0 for losing.) A setting is specified by a collection $\mathcal{X}$ of subsets of $\{1, 2, \ldots, n\}$; each such subset represent a collection of bidders that can simultaneously "win." For example, in a setting with $k$ copies of an item, where no bidder wants more than one copy, $\mathcal{X}$ would be all subsets of $\{1, 2, \ldots, n\}$ of cardinality at most $k$.

A generalization of this case, studied in the supplementary materials (Section 5), is *matroid* settings. These satisfy: (i) whenever $X \in \mathcal{X}$ and $Y \subseteq X, Y \in \mathcal{X}$; and (ii) for two sets $|I_1| < |I_2|, I_1, I_2 \in \mathcal{X}$, there is always an augmenting element $i_2 \in I_2 \setminus I_1$ such that $I_1 \cup \{i_2\} \in \mathcal{X}, \mathcal{X}$. The supplementary materials (Section 6) also consider arbitrary single-parameter settings, where the only assumption is that $\emptyset \in \mathcal{X}$. To ease comprehension, we often illustrate our main ideas using single-item auctions (where $\mathcal{X}$ is the singletons and the empty set).

We assume bidders' valuations are drawn from the continuous joint cumulative distribution $F$. Except in the extension in Section 4, we assume that the support of $F$ is limited to $[1, H]^n$. As in most of optimal auction theory [18], we usually assume that $F$ is a product distribution, with $F = F_1 \times F_2 \times \ldots \times F_n$ and each $v_i \sim F_i$ drawn independently but not identically. The *virtual*

*value* of bidder $i$ is denoted by $\phi_i(v_i) = v_i - \frac{1-F_i(v_i)}{f_i(v_i)}$. A distribution satisfies the *monotone-hazard rate (MHR) condition* if $f_i(v_i)/(1 - F_i(v_i))$ is nondecreasing; intuitively, if its tails are no heavier than those of an exponential distribution. In a fundamental paper, [18] proved that when every virtual valuation function is nondecreasing (the "regular" case), the auction that maximizes expected revenue for $n$ Bayesian bidders chooses winners in a way which maximizes the sum of the virtual values of the winners. This auction is known as Myerson's auction, which we refer to as $\mathcal{M}$. The result can be extended to the general, "non-regular" case by replacing the virtual valuation functions by "ironed virtual valuation functions." The details are well-understood but technical; see Myerson [18] and Hartline [13] for details.

**Sample Complexity, VC Dimension, and the Pseudo-Dimension** This section reviews several well-known definitions from learning theory. Suppose there is some domain $\mathcal{Q}$, and let $c$ be some unknown target function $c : \mathcal{Q} \to \{0, 1\}$. Let $\mathcal{D}$ be an unknown distribution over $\mathcal{Q}$. We wish to understand how many labeled samples $(x, c(x))$, $x \sim \mathcal{D}$, are necessary and sufficient to be able to output a $\hat{c}$ which agrees with $c$ almost everywhere with respect to $\mathcal{D}$. The distribution-independent sample complexity of learning $c$ depends fundamentally on the "complexity" of the set of binary functions $\mathcal{C}$ from which we are choosing $\hat{c}$. We define the relevant complexity measure next.

Let $S$ be a set of $m$ samples from $\mathcal{Q}$. The set $S$ is said to be *shattered* by $\mathcal{C}$ if, for every subset $T \subseteq S$, there is some $c_T \in \mathcal{C}$ such that $c_T(x) = 1$ if $x \in T$ and $c_T(y) = 0$ if $y \notin T$. That is, ranging over all $c \in \mathcal{C}$ induces all $2^{|S|}$ possible projections onto $S$. The *VC dimension* of $\mathcal{C}$, denoted $\mathcal{VC}(\mathcal{C})$, is the size of the largest set $S$ that can be shattered by $\mathcal{C}$.

Let $\mathrm{err}_S(\hat{c}) = (\sum_{x \in S} |c(x) - \hat{c}(x)|)/|S|$ denote the empirical error of $\hat{c}$ on $S$, and let $\mathrm{err}(\hat{c}) = \mathbb{E}_{x \sim D}[|c(x) - \hat{c}(x)|]$ denote the true expected error of $\hat{c}$ with respect to $\mathcal{D}$. A key result from learning theory [23] is: for every distribution $\mathcal{D}$, a sample $S$ of size $\Omega(\epsilon^{-2}(\mathcal{VC}(\mathcal{C}) + \ln \frac{1}{\delta}))$ is sufficient to guarantee that $\mathrm{err}_S(\hat{c}) \in [\mathrm{err}(\hat{c}) - \epsilon, \mathrm{err}(\hat{c}) + \epsilon]$ for *every* $\hat{c} \in \mathcal{C}$ with probability $1 - \delta$. In this case, the error on the sample is close to the true error, simultaneously for every hypothesis in $\mathcal{C}$. In particular, choosing the hypothesis with the minimum sample error minimizes the true error, up to $2\epsilon$. We say $\mathcal{C}$ is $(\epsilon, \delta)$-*uniformly learnable with sample complexity* $m$ if, given a sample $S$ of size $m$, with probability $1 - \delta$, for all $c \in \mathcal{C}$, $|\mathrm{err}_S(c) - \mathrm{err}(c)| < \epsilon$: thus, any class $\mathcal{C}$ is $(\epsilon, \delta)$-uniformly learnable with $m = \Theta\left(\frac{1}{\epsilon^2}\left(\mathcal{VC}(\mathcal{C}) + \ln \frac{1}{\delta}\right)\right)$ samples. Conversely, for every learning algorithm $\mathcal{A}$ that uses fewer than $\frac{\mathcal{VC}(\mathcal{C})}{\epsilon}$ samples, there exists a distribution $\mathcal{D}'$ and a constant $q$ such that, with probability at least $q$, $\mathcal{A}$ outputs a hypothesis $\hat{c}' \in \mathcal{C}$ with $\mathrm{err}(\hat{c}') > \mathrm{err}(\hat{c}) + \frac{\epsilon}{2}$ for some $\hat{c} \in \mathcal{C}$. That is, the true error of the output hypothesis is more than $\frac{\epsilon}{2}$ larger the best hypothesis in the class.

To learn real-valued functions, we need a generalization of VC dimension (which concerns binary functions). The *pseudo-dimension* [19] does exactly this.[2] Formally, let $c : \mathcal{Q} \to [0, H]$ be a real-valued function over $\mathcal{Q}$, and $\mathcal{C}$ the class we are learning over. Let $S$ be a sample drawn from $\mathcal{D}$, $|S| = m$, labeled according to $c$. Both the empirical and true error of a hypothesis $\hat{c}$ are defined as before, though $|\hat{c}(x) - c(x)|$ can now take on values in $[0, H]$ rather than in $\{0, 1\}$. Let $(r^1, \ldots, r^m) \in [0, H]^m$ be a set of *targets* for $S$. We say $(r^1, \ldots, r^m)$ *witnesses* the shattering of $S$ by $\mathcal{C}$ if, for each $T \subseteq S$, there exists some $c_T \in \mathcal{C}$ such that $f_T(x^i) \geq r^i$ for all $x^i \in T$ and $c_T(x^i) < r^i$ for all $x^i \notin T$. If there exists some $\vec{r}$ witnessing the shattering of $S$, we say $S$ is *shatterable* by $\mathcal{C}$. The *pseudo-dimension* of $\mathcal{C}$, denoted $\mathrm{d}_{\mathcal{C}}$, is the size of the largest set $S$ which is shatterable by $\mathcal{C}$. The sample complexity upper bounds of this paper are derived from the following theorem, which states that the distribution-independent sample complexity of learning over a class of real-valued functions $\mathcal{C}$ is governed by the class's pseudo-dimension.

**Theorem 2.1** *[E.g. [1]] Suppose $\mathcal{C}$ is a class of real-valued functions with range in $[0, H]$ and pseudo-dimension $\mathrm{d}_{\mathcal{C}}$. For every $\epsilon > 0, \delta \in [0, 1]$, the sample complexity of $(\epsilon, \delta)$-uniformly learning $f$ with respect to $\mathcal{C}$ is $m = O\left(\left(\frac{H}{\epsilon}\right)^2 \left(\mathrm{d}_{\mathcal{C}} \ln\left(\frac{H}{\epsilon}\right) + \ln\left(\frac{1}{\delta}\right)\right)\right).$*

Moreover, the guarantee in Theorem 2.1 is realized by the learning algorithm that simply outputs the function $c \in \mathcal{C}$ with the smallest empirical error on the sample.

**Applying Pseudo-Dimension to Auction Classes**    For the remainder of this paper, we consider classes of truthful auctions $\mathcal{C}$.[3] When we discuss some auction $c \in \mathcal{C}$, we treat $c : [0, H]^n \to \mathbb{R}$ as the function that maps (truthful) bid tuples to the revenue achieved on them by the auction $c$. Then, rather than minimizing error, we aim to maximize revenue. In our setting, the guarantee of Theorem 2.1 directly implies that, with probability at least $1 - \delta$ (over the $m$ samples), the output of the *empirical revenue maximization* learning algorithm — which returns the auction $c \in \mathcal{C}$ with the highest average revenue on the samples — chooses an auction with expected revenue (over the true underlying distribution $F$) that is within an additive $\epsilon$ of the maximum possible.

# 3    Single-Item Auctions

To illustrate out ideas, we first focus on single-item auctions. The results of this section are generalized significantly in the supplementary (see Sections 5 and 6).

Section 3.1 defines the class of $t$-level single-item auctions, gives an example, and interprets the auctions as approximations to virtual welfare maximizers. Section 3.2 proves that the pseudo-dimension of the set of such auctions is $O(nt \log nt)$, which by Theorem 2.1 implies a sample-complexity upper bound. Section 3.3 proves that taking $t = \Omega(\frac{H}{\epsilon})$ yields low representation error.

## 3.1    $t$-Level Auctions: The Single-Item Case

We now introduce $t$-*level auctions*, or $\mathcal{C}_t$ for short. Intuitively, one can think of each bidder as facing one of $t$ possible prices; the price they face depends upon the values of the other bidders. Consider, for each bidder $i$, $t$ numbers $0 \leq \ell_{i,0} \leq \ell_{i,1} \leq \ldots \leq \ell_{i,t-1}$. We refer to these $t$ numbers as *thresholds*. This set of $tn$ numbers defines a $t$-level auction with the following allocation rule. Consider a valuation tuple $\mathbf{v}$:

1. For each bidder $i$, let $t_i(v_i)$ denote the index $\tau$ of the largest threshold $\ell_{i,\tau}$ that lower bounds $v_i$ (or -1 if $v_i < \ell_{i,0}$). We call $t_i(v_i)$ the *level* of bidder $i$.
2. Sort the bidders from highest level to lowest level and, within a level, use a fixed lexicographical tie-breaking ordering $\succ$ to pick the winner.[4]
3. Award the item to the first bidder in this sorted order (unless $t_i = -1$ for every bidder $i$, in which case there is no sale).

The payment rule is the unique one that renders truthful bidding a dominant strategy and charges 0 to losing bidders — the winning bidder pays the lowest bid at which she would continue to win. It is important for us to understand this payment rule in detail; there are three interesting cases. Suppose bidder $i$ is the winner. In the first case, $i$ is the only bidder who might be allocated the item (other bidders have level -1), in which case her bid must be at least her lowest threshold. In the second case, there are multiple bidders at her level, so she must bid high enough to be at her level (and, since ties are broken lexicographically, this is her threshold to win). In the final case, she need not compete at her level: she can choose to either pay one level above her competition (in which case her position in the tie-breaking ordering does not matter) or she can bid at the same level as her highest-level competitors (in which case she only wins if she dominates all of those bidders at the next-highest level according to $\succ$). Formally, the payment $p$ of the winner $i$ (if any) is as follows. Let $\bar{\tau}$ denote the highest level $\tau$ such that there at least two bidders at or above level $\tau$, and $I$ be the set of bidders other than $i$ whose level is at least $\bar{\tau}$.

Monop    If $\bar{\tau} = -1$, then $p_i = \ell_{i,0}$ (she is the only potential winner, but must have level $\geq 0$ to win).
   Mult    If $t_i(v_i) = \bar{\tau}$ then $p_i = \ell_{i,\bar{\tau}}$ (she needs to be at level $\bar{\tau}$).

Unique If $t_i(v_i) > \bar{\tau}$, if $i \succ i'$ for all $i' \in I$, she pays $p_i = \ell_{i,\bar{\tau}}$, otherwise she pays $p_i = \ell_{i,\bar{\tau}+1}$ (she either needs to be at level $\bar{\tau} + 1$, in which case her position in $\succ$ does not matter, or at level $\bar{\tau}$, in which case she would need to be the highest according to $\succ$).

We now describe a particular $t$-level auction, and demonstrate each case of the payment rule.

**Example 3.1** Consider the following 4-level auction for bidders $a, b, c$. Let $\ell_{a,\cdot} = [2, 4, 6, 8]$, $\ell_{b,\cdot} = [1.5, 5, 9, 10]$, and $\ell_{c,\cdot} = [1.7, 3.9, 6, 7]$. For example, if bidder $a$ bids less than 2 she is at level $-1$, a bid in $[2, 4)$ puts her at level 0, a bid in $[4, 6)$ at level 1, a bid in $[6, 8)$ at level 2, and a bid of at least 8 at level 3. Let $a \succ b \succ c$.

Monop If $v_a = 3, v_b < 1.5, v_c < 1.7$, then $b, c$ are at level $-1$ (to which the item is never allocated). So, $a$ wins and pays 2, the minimum she needs to bid to be at level 0.

Mult If $v_a \geq 8, v_b \geq 10, v_c < 7$, then $a$ and $b$ are both at level 3, and $a \succ b$, so $a$ will win and pays 8 (the minimum she needs to bid to be at level 3).

Unique If $v_a \geq 8, v_b \in [5, 9], v_c \in [3.9, 6]$, then $a$ is at level 3, and $b$ and $c$ are at level 1. Since $a \succ b$ and $a \succ c$, $a$ need only pay 4 (enough to be at level 1). If, on the other hand, $v_a \in [4, 6], v_b = [5, 9]$ and $v_c \geq 6$, $c$ has level at least 2 (while $a, b$ have level 1), but $c$ needs to pay 6 since $a, b \succ c$.

**Remark 3.2 (Connection to virtual valuation functions)** $t$-level auctions are naturally interpreted as discrete approximations to virtual welfare maximizers, and our representation error bound in Theorem 3.4 makes this precise. Each level corresponds to a constraint of the form "If any bidder has level at least $\tau$, do not sell to any bidder with level less than $\tau$." We can interpret the $\ell_{i,\tau}$'s (with fixed $\tau$, ranging over bidders $i$) as the bidder values that map to some common virtual value. For example, 1-level auctions treat all values below the single threshold as having negative virtual value, and above the threshold uses values as proxies for virtual values. 2-level auctions use the second threshold to the refine virtual value estimates, and so on. With this interpretation, it is intuitively clear that as $t \to \infty$, it is possible to estimate bidders' virtual valuation functions and thus approximate Myerson's optimal auction to arbitrary accuracy.

## 3.2 The Pseudo-Dimension of $t$-Level Auctions

This section shows that the pseudo-dimension of the class of $t$-level single-item auctions with $n$ bidders is $O(nt \log nt)$. Combining this with Theorem 2.1 immediately yields sample complexity bounds (parameterized by $t$) for learning the best such auction from samples.

**Theorem 3.3** *For a fixed tie-breaking order, the set of $n$-bidder single-item $t$-level auctions has pseudo-dimension $O(nt \log(nt))$.*

*Proof:* Recall from Section 2 that we need to upper bound the size of every set that is shatterable using $t$-level auctions. Fix a set of samples $S = (\mathbf{v^1}, \dots, \mathbf{v^m})$ of size $m$ and a potential witness $R = (r^1, \dots, r^m)$. Each auction $c$ induces a binary labeling of the samples $\mathbf{v}^j$ of $S$ (whether $c$'s revenue on $\mathbf{v}^j$ is at least $r^j$ or strictly less than $r^j$). The set $S$ is shattered with witness $R$ if and only if the number of distinct labelings of $S$ given by any $t$-level auction is $2^m$.

We upper-bound the number of distinct labelings of $S$ given by $t$-level auctions (for some fixed potential witness $R$), counting the labelings in two stages. Note that $S$ involves $nm$ numbers — one value $v_i^j$ for each bidder for each sample, and a $t$-level auction involves $nt$ numbers — $t$ thresholds $\ell_{i,\tau}$ for each bidder. Call two $t$-level auctions with thresholds $\{\ell_{i,\tau}\}$ and $\{\hat{\ell}_{i,\tau}\}$ *equivalent* if

1. The relative order of the $\ell_{i,\tau}$'s agrees with that of the $\hat{\ell}_{i,\tau}$'s, in that both induce the same permutation of $\{1, 2, \dots, n\} \times \{0, 1, \dots, t-1\}$.
2. merging the sorted list of the $v_i^j$'s with the sorted list of the $\ell_{i,\tau}$'s yields the same partition of the $v_i^j$'s as does merging it with the sorted list of the $\hat{\ell}_{i,\tau}$'s.

Note that this is an equivalence relation. If two $t$-level auctions are equivalent, every comparison between a valuation and a threshold or two valuations is resolved identically by those auctions.

Using the defining properties of equivalence, a crude upper bound on the number of equivalence classes is

$$(nt)! \cdot \binom{nm+nt}{nt} \leq (nm+nt)^{2nt}. \tag{1}$$

We now upper-bound the number of distinct labelings of $S$ that can be generated by $t$-level auctions in a single equivalence class $C$. First, as all comparisons between two numbers (valuations or thresholds) are resolved identically for all auctions in $C$, each bidder $i$ in each sample $\mathbf{v}^j$ of $S$ is assigned the same level (across auctions in $C$), and the winner (if any) in each sample $\mathbf{v}^j$ is constant across all of $C$. By the same reasoning, the identity of the parameter that gives the winner's payment (some $\ell_{i,\tau}$) is uniquely determined by pairwise comparisons (recall Section 3.1) and hence is common across all auctions in $C$. The payments $\ell_{i,\tau}$, however, can vary across auctions in the equivalence class.

For a bidder $i$ and level $\tau \in \{0, 1, 2, \ldots, t-1\}$, let $S_{i,\tau} \subseteq S$ be the subset of samples in which bidder $i$ wins and pays $\ell_{i,\tau}$. The revenue obtained by each auction in $C$ on a sample of $S_{i,\tau}$ is simply $\ell_{i,\tau}$ (and independent of all other parameters of the auction). Thus, ranging over all $t$-level auctions in $C$ generates at most $|S_{i,\tau}|$ distinct binary labelings of $S_{i,\tau}$ — the possible subsets of $S_{i,\tau}$ for which an auction meets the corresponding target $r^j$ form a nested collection.

Summarizing, within the equivalence class $C$ of $t$-level auctions, varying a parameter $\ell_{i,\tau}$ generates at most $|S_{i,\tau}|$ different labelings of the samples $S_{i,\tau}$ and has no effect on the other samples. Since the subsets $\{S_{i,\tau}\}_{i,\tau}$ are disjoint, varying all of the $\ell_{i,\tau}$'s (i.e., ranging over $C$) generates at most

$$\prod_{i=1}^{n} \prod_{\tau=0}^{t-1} |S_{i,\tau}| \leq m^{nt} \tag{2}$$

distinct labelings of $S$.

Combining (1) and (2), the class of all $t$-level auctions produces at most $(nm + nt)^{3nt}$ distinct labelings of $S$. Since shattering $S$ requires $2^m$ distinct labelings, we conclude that $2^m \leq (nm + nt)^{3nt}$, implying $m = O(nt \log nt)$ as claimed. ∎

### 3.3 The Representation Error of Single-Item $t$-Level Auctions

In this section, we show that for every bounded product distribution, there exists a $t$-level auction with expected revenue close to that of the optimal single-item auction when bidders are independent and bounded. The analsysis "rounds" an optimal auction to a $t$-level auction without losing much expected revenue. This is done using thresholds to approximate each bidder's virtual value: the lowest threshold at the bidder's monopoly reserve price, the next $\frac{1}{\epsilon}$ thresholds at the values at which bidder $i$'s virtual value surpasses multiples of $\epsilon$, and the remaining thresholds at those values where bidder $i$'s virtual value reaches powers of $1 + \epsilon$. Theorem 3.4 formalizes this intuition.

**Theorem 3.4** *Suppose $F$ is distribution over $[1, H]^n$. If $t = \Omega\left(\frac{1}{\epsilon} + \log_{1+\epsilon} H\right)$, $\mathcal{C}_t$ contains a single-item auction with expected revenue at least $1 - \epsilon$ times the optimal expected revenue.*

Theorem 3.4 follows immediately from the following lemma, with $\alpha = \gamma = 1$. We prove this more general result for later use.

**Lemma 3.5** *Consider $n$ bidders with valuations in $[0, H]$ and with $\mathbb{P}[\max_i v_i > \alpha] \geq \gamma$. Then, $\mathcal{C}_t$ contains a single-item auction with expected revenue at least a $1 - \epsilon$ times that of an optimal auction, for $t = \Theta\left(\frac{1}{\gamma\epsilon} + \log_{1+\epsilon} \frac{H}{\alpha}\right)$.*

*Proof:* Consider a fixed bidder $i$. We define $t$ thresholds for $i$, bucketing $i$ by her virtual value, and prove that the $t$-level auction $\mathcal{A}$ using these thresholds for each bidder closely approximates the expected revenue of the optimal auction $\mathcal{M}$. Let $\epsilon'$ be a parameter defined later.

Set $\ell_{i,0} = \phi_i^{-1}(0)$, bidder $i$'s monopoly reserve.[5] For $\tau \in [1, \lceil \frac{1}{\gamma \epsilon'} \rceil]$, let $\ell_{i,\tau} = \phi_i^{-1}(\tau \cdot \alpha \gamma \epsilon')$ ($\phi_i \in [0, \alpha]$). For $\tau \in [\lceil \frac{1}{\gamma \epsilon'} \rceil, \lceil \frac{1}{\gamma \epsilon'} \rceil + \lceil \log_{1 + \frac{\epsilon}{2}} \frac{H}{\alpha} \rceil]$, let $\ell_{i,\tau} = \phi_i^{-1}(\alpha(1 + \frac{\epsilon}{2})^{\tau - \lceil \frac{1}{\gamma \epsilon'} \rceil})$ ($\phi_i > \alpha$).

Consider a fixed valuation profile $\mathbf{v}$. Let $i^*$ denote the winner according to $\mathcal{A}$, and $i'$ the winner according to the optimal auction $\mathcal{M}$. If there is no winner, we interpret $\phi_{i^*}(v_{i^*})$ and $\phi_{i'}(v_{i'})$ as 0. Recall that $\mathcal{M}$ always awards the item to a bidder with the highest positive virtual value (or no one, if no such bidders exist). The definition of the thresholds immediately implies the following.

1. $\mathcal{A}$ only allocates to non-negative ironed virtual-valued bidders.
2. If there is no tie (that is, there is a unique bidder at the highest level), then $i' = i^*$.
3. When there is a tie at level $\tau$, the virtual value of the winner of $\mathcal{A}$ is close to that of $\mathcal{M}$:
   If $\tau \in [0, \lceil \frac{1}{\gamma \epsilon'} \rceil]$ then $\phi_{i'}(v_{i'}) - \phi_{i^*}(v_{i^*}) \leq \alpha \gamma \epsilon'$;

   if $\tau \in [\lceil \frac{1}{\gamma \epsilon'} \rceil, \lceil \frac{1}{\gamma \epsilon'} \rceil + \lceil \log_{1 + \frac{\epsilon}{2}} \frac{H}{\alpha} \rceil]$, $\frac{\phi_{i^*}(v_{i^*})}{\phi_{i'}(v_{i'})} \geq 1 - \frac{\epsilon}{2}$.

These facts imply that

$$\mathbb{E}_{\mathbf{v}}[\text{Rev}(\mathcal{A})] = \mathbb{E}_{\mathbf{v}}[\phi_{i^*}(v_{i^*})] \geq (1 - \frac{\epsilon}{2}) \cdot \mathbb{E}_{\mathbf{v}}[\phi_{i'}(v_{i'})] - \alpha \gamma \epsilon' = (1 - \frac{\epsilon}{2}) \cdot \mathbb{E}_{\mathbf{v}}[\text{Rev}(\mathcal{M})] - \alpha \gamma \epsilon'. \quad (3)$$

are equal. The first and final equality follow from $\mathcal{A}$ and $\mathcal{M}$'s allocations depending on ironed virtual values, not on the values themselves, thus, the ironed virtual values are equal in expectation to the unironed virtual values, and thus the revenue of the mechanisms (see [13], Chapter 3.5 for discussion).

As $\mathbb{P}[\max_i v_i > \alpha] \geq \gamma$, it must be that $\mathbb{E}[\text{Rev}(\mathcal{M})] \geq \alpha \gamma$ (a posted price of $\alpha$ will achieve this revenue). Combining this with (3), and setting $\epsilon' = \frac{\epsilon}{2}$ implies $\mathbb{E}_{\mathbf{v}}[\text{Rev}(\mathcal{A})] \geq (1 - \epsilon) \mathbb{E}_{\mathbf{v}}[\text{Rev}(\mathcal{M})]$. ∎

Combining Theorems 2.1 and 3.4 yields the following Corollary 3.6.

**Corollary 3.6** *Let $F$ be a product distribution with all bidders' valuations in $[1, H]$. Assume that $t = \Theta\left(\frac{1}{\epsilon} + \log_{1 + \epsilon} H\right)$ and $m = O\left(\left(\frac{H}{\epsilon}\right)^2 \left(nt \log(nt) \log \frac{H}{\epsilon} + \log \frac{1}{\delta}\right)\right) = \tilde{O}\left(\frac{H^2 n}{\epsilon^3}\right)$. Then with probability at least $1 - \delta$, the single-item empirical revenue maximizer of $\mathcal{C}_t$ on a set of $m$ samples from $F$ has expected revenue at least $1 - \epsilon$ times that of the optimal auction.*

## Open Questions

There are some significant opportunities for follow-up research. First, there is much to do on the design of *computationally efficient* (in addition to sample-efficient) algorithms for learning a near-optimal auction. The present work focuses on sample complexity, and our learning algorithms are generally not computationally efficient.[6] The general research agenda here is to identify auction classes $\mathcal{C}$ for various settings such that:

1. $\mathcal{C}$ has low representation error;
2. $\mathcal{C}$ has small pseudo-dimension;
3. There is a polynomial-time algorithm to find an approximately revenue-maximizing auction from $\mathcal{C}$ on a given set of samples.[7]

There are also interesting open questions on the statistical side, notably for multi-parameter problems. While the negative result in [11] rules out a universally good upper bound on the sample complexity of learning a near-optimal mechanism in multi-parameter settings, we suspect that positive results are possible for several interesting special cases.

## Footnotes

[1]See Balcan et al. [3] for an extension to the case of a large finite supply.

[2]The *fat-shattering dimension* is a weaker condition that is also sufficient for sample complexity bounds. All of our arguments give the same upper bounds on the pseudo-dimension and the fat-shattering dimension of various auction classes, so we present the stronger statements.

[3] An auction is *truthful* if truthful bidding is a dominant strategy for every bidder. That is: for every bidder $i$, and all possible bids by the other bidders, $i$ maximizes its expected utility (value minus price paid) by bidding its true value. In the single-parameter settings that we study, the expected revenue of the optimal non-truthful auction (measured at a Bayes-Nash equilibrium with respect to the prior distribution) is no larger than that of the optimal truthful auction.

[4] When the valuation distributions are regular, this tie-breaking can be done by value, or randomly; when it is done by value, this equates to a generalization of VCG with nonanonymous reserves (and is IC and has identical representation error as this analysis when bidders are regular).

[5]Recall from Section 2 that $\phi_i$ denotes the virtual valuation function of bidder $i$. (From here on, we always mean the ironed version of virtual values.) It is convenient to assume that these functions are strictly increasing (not just nondecreasing); this can be enforced at the cost of losing an arbitrarily small amount of revenue.

[6]There is a clear parallel with *computational* learning theory [22]: while the information-theoretic foundations of classification (VC dimension, etc. [23]) have been long understood, this research area strives to understand which low-dimensional concept classes are learnable in polynomial time.

[7]The sample-complexity and performance bounds implied by pseudo-dimension analysis, as in Theorem 2.1, hold with such an approximation algorithm, with the algorithm's approximation factor carrying through to the learning algorithm's guarantee. See also [4, 11].

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
