[Reviews · NeurIPS 2015]

Submitted by Assigned_Reviewer_1

The paper is concerned with design of near optimal auction mechanisms where optimality is measured with respect to revenue. Auction theory has characterized these auctions as ones that assign the object to the bidder with highest virtual valuation. Nevertheless, this requires knowledge of the distribution F_i from which valuation v_i is drawn from. Moreover, even with this knowledge the payment rule for these auctions can be extremely complicated to be used in practice. Instead, the authors propose a simple family of auctions dubbed t-level auctions which admit a simple payment rule. Moreover, in order to deal with the problem of not knowing

F_i, the authors propose doing empirical revenue maximization over this family of auctions.

The main contribution of this work is showing that by letting t = O(\frac{1}{\eps})

and with a sample size in O(1/\eps^3), the empirical revenue maximizing t-level auction achieves revenue that is (1 - \eps)-optimal.

The paper introduces an interesting connection between learning theory and auction theory. Recently this connection has been explored by several people and the authors provide a good review of this previous work. The paper is in general well written and although the concepts and proofs are basic for both machine learning and auction theory, it is an interesting connection that I believe should be explored more by the learning community.

Here are some comments to improve the paper: (1) The explanation on pseudo-dimension is too long. Given that this a learning conference I believe that a simple reference to the definition is enough as most people know what this is.

(2) The explanation of equivalence classes is not entirely clear to me. Why most the levels induce the same permuation? It seems that the second condition is good enough.

(3) In the proof of lemma 3.5 I believe you mean \phi \in [0, \alpha] (4) It would be good to put in the appendix the reasoning behind equation (3) since the fact that expected revenue equals expected allocation of virtual valuations might not be a standard fact in the learning community.

(5) First paragraph of the appendix : " bounded case, following ideas from This... " I am sure this sentence shouldn't be here.
Summary: This paper presents an interesting approach to the study of optimal mechanism design via the use of machine learning techniques.

Results are correct and present an interesting connection between auction theory and machine learning.

Submitted by Assigned_Reviewer_2

This paper addresses the problem of learning reserve prices that approximately maximize revenue, using sample draws from an unknown distribution over bidder valuations. The authors introduce t-level auctions, in which (roughly speaking) each bidder's bid space is effectively discretized into levels, and the bidder whose bid falls on the highest level wins and pays the lowest value that falls on its lowest level required to win.

The authors bound the number of samples needed to find an approximately revenue-maximizing auction from all auctions in a set C (e.g., from the set of 10-level auctions). They bound the difference in revenue between the revenue-maximizing t-level auction and the optimal auction. Results are presented for single-item auctions but are generalized to matroid settings and single-parameter settings.

This paper is well-written and does a good job of comparing/contrasting the problem, approach, and results to other relevant papers in the literature. The work is novel and makes connections across multiple disciples (auction theory and learning theory). The paper is of very strong technical quality, with some of the most interesting technical results extending into the appendix.

One issue that might be of practical importance: after a particular t-level auction is implemented, there is likely information lost about the subsequent bidder valuations. While truth-telling is a dominant strategy, so is reporting the lowest value that falls on the same level as the bidder's true value. If the bidder is concerned about revealing information to the auctioneer (e.g., it thinks the bidder may try to re-learn the optimal reserves in a subsequent period, in the case that F is non-stationary), it seems reasonable to think that the bidder would choose this "lowest value on interval" strategy, since it gets the same payment and allocation as truthfully reporting but reveals less information to the auctioneer. If the auctioneer wants to re-learn optimal reserves in the future or otherwise understand how bidders will respond to a change in the mechanism, losing some precision in observed bidder valuations could be problematic.

As a more fundamental side-note: I understand the theoretical importance in any case, so this point does not really impact my score, but I am a little skeptical of the value of revenue-maximizing auctions in practice, at least in some well-studied real-world domains, because they don't capture the fact that bidders are often making decisions over time, and extracting as much revenue as possible in the current auction doesn't necessarily coincide with maximizing long-term revenue. I would be curious to hear any normative statement from the authors (in either direction) about the practical importance of optimal auctions in real-world domains, or how to otherwise reconcile this tension in optimizing short- versus long-term revenue. If there is space and a good argument to be made, adding a comment to the paper would be valuable.

Minor question: in a footnote it is mentioned that results hold for other tie-breaking rules. Which ones? It may be worth calling out whether results hold when ties are broken uniformly at random (my initial thought is that results would not hold, but the authors probably have a definitive answer).
Summary: This paper introduces t-level auctions, quantifies how much data is needed to learn a near-optimal t-level auction, and further quantifies how far the optimal t-level auction is from the optimal auction. The paper is well-written, of strong technical quality, and contains novel contributions joining work across disciplines.

Submitted by Assigned_Reviewer_3

The paper considers the problem of designing revenue-maximizing auctions given samples from the "valuation distribution" F of the users.

The paper shows an upper bound on the number of samples required to obtain an auction whose performance is close to an optimal auction.

The key ingredient is to design a class of auctions C_t, called t-level auctions. As t grows one of the auctios in C_t is close to the optimal auction. This is interesting since t-level functions have only n*t free parameters, and hence applying basic learning theoretic arguments, the authors provide an upper bound on the number of samples required to learn a good auction. While the existence of such auctions are interesting to a game-theoretic audience, I am slightly skeptical about the interest it can generate in NIPS. I think there are a number of interesting problems such as the one the authors consider here at the intersection of learning and game theory, however, they need to make a better case, for example with either faster algorithms, or showing some special classes of distributions F for which learning the auction is possible much faster, and or with much fewer samples. Finally, it would be interesting to know what kind of lower bounds exist on the number of samples required to learn a near-optimal auction. In particular, since in many learning problems the number of samples is close to the logarithm of the covering number, it would be interesting to see if some such results hold in this setting.

The paper is well written, although a little more motivation might be helpful for someone not completely knowledgeable about the auction theory literature.
Summary: The main contribution is showing that there exist a class of auctions C1< C2< ... such that as t grows Ct contains an auction that is nearly optimal. Even though i like this result, the learning arguments are straight-forward, and are existential (computationally hard). I also believe the results are slightly incremental in that sense. One plus of this paper at nips could be introducing these interesting problems in algorithmic game theory to the learning community in a learning framework.

Author Feedback
Author rebuttal: We thank the reviewers for their careful reviews.

Reviewer 2:
We agree that the second condition of the definition of equivalence classes is enough for the proof, and thank you for catching our other typos.

Reviewer 3:
Tie-breaking at random works for arbitrary (regular or irregular) bidders; if bidders are regular, one can also tie-break according to value (e.g., pick the bidder with the highest level, breaking ties amongst those with the same level by choosing the one with largest bid).

One way to fix the issue you point out about bidders "rounding down" their bids being a weakly dominant strategy is to break ties by value (if bidders are regular): if two bidders have some level \tau which overlaps somewhat in value-space, there will be values at that level for which it will no longer be weakly dominant to round down to the threshold value. This would occur, for example, if two bidders have sufficiently similar valuation distributions.

Reviewer 4:
We point out that remark 4.4 points out a known lower bound for learning near-optimal auctions: one needs sample complexity to grow at least linearly in n and 1/\epsilon. We agree that showing the connection to the covering number of the space of auctions would be an interesting direction to consider.